# Pragmatically Learning from Pedagogical Demonstrations in Multi-Goal Environments

**Hugo Caselles-Dupré, Olivier Sigaud, Mohamed Chetouani**
Sorbonne Université, CNRS, Institut des Systèmes Intelligents et de Robotique (ISIR)
Paris, France
casellesdupre.hugo@gmail.com,olivier.sigaud,mohamed.chetouani@isir.upmc.fr

## Abstract

Learning from demonstration methods usually leverage close to optimal demonstrations to accelerate training. By contrast, when demonstrating a task, human teachers deviate from optimal demonstrations and pedagogically modify their behavior by giving demonstrations that best disambiguate the goal they want to demonstrate. Analogously, human learners excel at pragmatically inferring the intent of the teacher, facilitating communication between the two agents. These mechanisms are critical in the few demonstrations regime, where inferring the goal is more difficult. In this paper, we implement pedagogy and pragmatism mechanisms by leveraging a Bayesian model of Goal Inference from demonstrations (BGI). We highlight the benefits of this model in multi-goal teacher-learner setups with two artificial agents that learn with goal-conditioned Reinforcement Learning. We show that combining BGI-agents (a pedagogical teacher and a pragmatic learner) results in faster learning and reduced goal ambiguity over standard learning from demonstrations, especially in the few demonstrations regime. We provide the code for our experiments [1], as well as an illustrative video explaining our approach [2].

## 1 Introduction

Imagine a teacher-learner setup where 3 colored blocks are lying on a table, the green and red blocks are close to each other and the blue block is further away. Assume the teacher wants to demonstrate how to have the blue block close to the green one (see Fig. 1). A naive teacher may move the blue block to the green one, but this would be ambiguous: is the goal to put the blue block close to the green one or to the red one? By contrast, a pedagogical teacher would move the green block close to the blue one, hence away from the red one, resolving the ambiguity as illustrated in Fig. 1.

As this example shows, in real life, when a teacher shows how to do something with a demonstration, the agent receiving the demonstration does not have access to the intended goal of the demonstration: it must infer this goal. In many situations, a single demonstration may be correctly interpreted as demonstration of a variety of goals – we call this goal ambiguity. In that case, the teacher may use pedagogy to help the learner infer the intended goal, and the learner may use pragmatism in order to increase its chance to infer the right goal. This is the situation we address in this paper.

To infer that a demonstration facilitates learning, the pedagogical teacher may ask herself what goal she would infer if she was the learner. Using her policy and Bayesian inference, she can determine the probability that she would infer the right goal given this demonstration, and assume the learner would do the same. Now, the learner has to interpret the demonstration to infer the goal. Given the same inference mechanism as the teacher, a literal learner would perform this inference using her

---

[1] https://github.com/Caselles/NeurIPS22-demonstrations-pedagogy-pragmatism
[2] https://youtu.be/V4n16IjkNyw

own policy. On top of that, a pragmatic learner could first train her policy to predict her own goals from her own trajectories, increasing her likelihood to correctly infer the teacher's goal.

Those mechanisms, pedagogy from the teacher and pragmatism from the learner, facilitate communication between the two agents and thus improve learning by reducing ambiguity about the inferred goal. Pedagogy and pragmatism are concepts borrowed from cognitive science research. On the one hand, pedagogy is defined as the optimization of teaching concepts from examples [8, 9]. On the other hand, pragmatism is a property used to resolve ambiguities of intention interpretation from teaching signals, which can be language with for instance the rational speech act (RSA) [34, 20], or actions with pedagogical demonstrations [38].

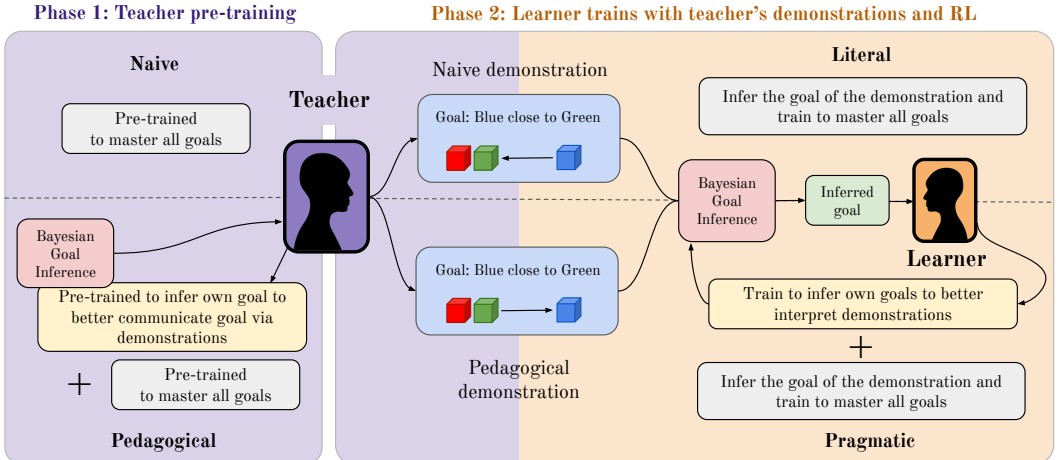

Fig. 1: Overview of our teacher-learner setup. All goal inferences are performed using Bayesian Goal Inference, an inference method that computes the goal probability from a demonstration using the agent's policy. In the first phase, the naive teacher is pre-trained to master all goals while the pedagogical teacher is additionally pre-trained to better infer goals from its own trajectories and effectively modifies its policy to produce demonstrations with less ambiguous goals. In the second phase, the learner is trained using the teacher's demonstrations and goal-conditioned Reinforcement Learning. The literal learner infers goals from demonstrations and is rewarded for correct inferences while the pragmatic learner additionally trains itself to infer its own goals from its own trajectories and effectively modifying its policy to better infer goals from the teacher's demonstrations.

Such ideas have been leveraged into the general theory of "Inferential Social Learning" (ISL) [22] in order to explain our abilities as learners and teachers to "interpret and generate evidence in social contexts". ISL is characterized as an inference mechanism guided by an intuitive understanding of how people think, plan, and act. It leverages Bayesian inference to explain several key aspects involved in pedagogy and pragmatism: pedagogy can help avoid goal ambiguity [26, 27] or accelerate goal inference from a learner (i.e. legibility) [29], and pragmatism can help better infer goals [14], or detect pedagogy [22], or generalize from induction [23] (what can I deduce from a teaching signal?). In this paper, we show that using Bayesian inference for disambiguating goals (pedagogy) and better infer goals (pragmatism) can improve the learning capabilities of artificial agents. In particular, we show that this mechanism is critical to reduce ambiguity when the teacher performs few demonstrations, which may contribute to more feasible robot teaching in the future.

We present an approach for implementing inferential mechanisms to improve learning from demonstrations in an artificial teacher-learner setup, as illustrated in Fig. 1. We use two multi-goal environments: a simple one for illustrating our point, and a more complex simulated robotics block manipulation environment to show the robustness of our approach. In both environments, we create naive teachers trained to master all goals, and pedagogical teachers trained to provide demonstrations that best disambiguate between all goals and thus facilitate goal inference for learners. We then create literal learners, which learn from teacher demonstrations, and pragmatic learners, which adapt their policy to better infer goals from the teacher's demonstrations. The mechanism for implementing pedagogy and pragmatism is an additional reward for correct goal inference of the agent's own trajectories. Goal inference is implemented using a Bayesian model and the agent's policy.

We make the following contributions:

1. Inspired by the ISL framework, we introduce pedagogical teaching and pragmatic learning mechanisms based on Bayesian Goal Inference and show that these mechanisms can be leveraged in a cooperative artificial teacher-learner training setup where both agents use goal-conditioned policies.

2. We show that pedagogy and pragmatism help reduce goal ambiguity and result in faster learning, which is especially helpful when the teacher performs fewer demonstrations.

## 2  Related work

Our work is related to several strongly connected areas.

**Bayesian inference.** Bayesian Inference was already used as a key mechanism for goal inference in the context of inverse planning [7, 5, 43]. It uses the Bayes formula to compute the probabilities of goals given the actions and the policy. In our work we use it as a tool to produce non-ambiguous demonstrations on the teacher side and pragmatic inference on the learner side.

**Pedagogical demonstrations.** In an attempt to explain how humans demonstrate tasks to each other, Ho et al. [29, 28, 26] introduce the Observer Belief MDP model and show that producing more pedagogical demonstrations results in better performance. However, their work does not investigate how this affects an artificial learner, nor the effect of pragmatism in the learner.

**Legibility in Human-Robot Interaction.** Similarly to us, [13, 32] modify robot action sequences to allow an observer (a human in their case) to more quickly and successfully understand a robot's goal from a more legible trajectory. In our work, Bayesian goal inference is employed in pedagogical teachers to allow them to generate less ambiguous goal demonstration using additional reward for correct goal inference on their own motion.

**Pragmatic inference.** A long line of work in linguistics, natural language processing, and cognitive science has studied pragmatics: how linguistic meaning is affected by context and communicative goals [18, 17, 21]. Another line of work on pragmatics, more related to our work, focuses on inferring meaning from action in a context. It has been applied to Robotics [33, 15], where a pragmatic robot better infers the objective of a teacher by considering its pedagogical intentions, in a multi-agent game theory context. In our case, we are interested in how a pragmatic learner can learn by itself to better infer the goals of a demonstrator, resulting in faster task learning, which is different from detecting if the teacher is pedagogical or not.

**Demonstrations in goal-conditioned tasks.** Our learning from demonstration mechanism is related to work on learning from demonstrations in multi-goal environments, by contrast with the more common setup where the agent only learns one task. Most approaches targeting fewer demonstrations address the one task setup [1, 44, 42, 40]. Approaches using demonstrations for goal-conditioned tasks such as GAIL [25], DCRL [10] and CLIC [16] combine goal-conditioned RL and imitation learning with additional rewards. In our work, we also combine a goal-conditioned RL algorithm (GANGSTR [3]) and a slightly modified version of an algorithm learning both from demonstration and from the agent's own experience (SQIL [24]). The specificity of our pipeline is that we can easily improve it with pedagogical demonstrations and pragmatic inference to accelerate training, but in principle all goal-conditioned RL algorithms using demonstrations could be improved in such a way.

## 3  Methods

We consider multi-goal environments with teacher-learner scenarios where the learner is both trained with teachers' demonstrations and using its own exploration. The teacher $T$ and the learner $L$ are both represented by their respective goal-conditioned policies $\pi_T(.|g)$ and $\pi_L(.|g)$. These policies might be learned using any algorithm (multi-armed bandits, RL, evolution strategies, etc.), here we use a goal-conditioned RL (GCRL) algorithm [11] called GANGSTR [3], which we describe in our experimental setup. Our multi-goal environments present goal ambiguity, as the same trajectory can simultaneously reach at least two goals $(g_1, g_2)$. Considering this hypothesis, it is generally not possible to reliably predict the pursued goal from a demonstration reaching both $g_1$ and $g_2$.

Both agents share common goal, state and action spaces. They communicate through teacher's demonstrations, learner's inferred goal and teacher feedback on this inference. To efficiently exploit these signals, we introduce a Bayesian Goal Inference (BGI) mechanism helping agents infer goals from demonstrations. The teacher uses it to generate less ambiguous demonstrations (pedagogy), while the learner uses it to infer the teacher's goal from the demonstration (pragmatism). We refer to the pedagogical teacher and pragmatic learners as BGI-agents.

The training process is spread across two phases: 1) the teacher is pre-trained to master all goals in the environment using GCRL, and 2) the learner infers the goal from a teacher's demonstration using Bayesian Goal Inference, the teacher provides feedback on the inference, and the learner is rewarded for correct predictions. The learner then attempts at reaching the inferred goal and iteratively improves its policy using GCRL.

Below we formally define BGI, then we define naive teachers and literal learners, which do not leverage the full benefits of BGI. Finally we introduce pedagogical teachers and pragmatic learners, which use BGI to facilitate communication with each other.

### 3.1 Bayesian Goal Inference in Teacher-Learner Interactions

We formally introduce the BGI mechanism used to infer goals from a goal-conditioned policy. In our work, 1) the teacher uses it to implement pedagogy, 2) the learner uses it to infer goals from the demonstrations of the teacher, and 3) the learner uses it to implement pragmatism.

**Inferring the Goal from a Demonstration** In a goal-conditioned Markov environment, the probability of observing a demonstration $d = ((s_1, a_1), .., (s_n, a_n))$ given a goal $g$ and the goal-conditioned policy $\pi(.|g)$ that generated it can be written [26, 6] as:

$$
\begin{aligned}
\mathbb{P}(d|g) &= \mathbb{P}((s_1, a_1), .., (s_n, a_n)|g) \\
&= \prod_{i=1}^{n} \pi(a_i|s_i, g) \cdot \mathbb{P}(s_{i+1}|s_i, a_i) = \prod_{i=1}^{n} \pi(a_i|s_i, g),
\end{aligned}
\tag{1}
$$

where $\mathbb{P}(s_{i+1}|s_i, a_i) = 1$ as we consider a deterministic environment.

Now, to infer a goal given a demonstration, by using Bayes' rule we can derive $\mathbb{P}(G|d)$, the probability distribution over the goal space $G$ given the demonstration:

$$
\mathbb{P}(G|d) \propto \mathbb{P}(d|G) \cdot \mathbb{P}(G) = \prod_{i=1}^{n} \pi(a_i|s_i, G) \cdot \mathbb{P}(G).
\tag{2}
$$

For each goal $g$, the prior $\mathbb{P}(G)$ is uniform if not specified otherwise. Given $\mathbb{P}(G|d)$, an agent can infer the goal of a demonstration by either taking the most probable goal, or sampling from the distribution. To perform this inference, the agent uses its own policy.

**Learning Goal Inference from Own Trajectories** When playing its policy, an agent produces trajectories which we call demonstrations when they are produced for other agents. A GCRL agent can leverage the BGI mechanism on its own trajectories to improve its ability to infer goals. To do so, during training, the agent selects a goal, performs a trajectory, infers the goal from the trajectory, and rewards itself if the inference is correct. This reinforces the policy towards actions that lead to better goal inference. We use this to implement both pedagogy in the teacher and pragmatism in the learner.

### 3.2 Naive/Pedagogical Teacher and Literal/Pragmatic Learner training

We now present the two training phases in our teacher-learner setup. First, the teacher is pre-trained, and then it provides demonstrations for the learner to train with. The two-phases training process is presented in Algorithm 1.

---

**Algorithm 1** Two-phases training of the teacher and the learner

---

1: **PHASE 1: Teacher pre-training** ($\pi_T(.|g)$)
2: Initialize $\pi_T(.|g)$, goal set $G_T$ to {first goal} and pedagogical Boolean variable
3: **for** $epoch = 1, 2, \ldots$ **do**
4:    Randomly sample goal $g$ from goal set $G_T$
5:    Run policy $\pi_T(.|g)$, obtain trajectory $t = ((s_1, a_1, r_1), .., (s_n, a_n, r_n))$ and achieved goal $g_a$
6:    **if** pedagogical Boolean $= 1$ and $g = g_a$ **then**
7:       Infer own goal $\widehat{g}$ with BGI on $t$: if $\widehat{g} = g$, add pedagogical reward to $t$
8:    **end if**
9:    Add $t$ to replay buffer and if $g_a$ is new, add $g_a$ to goal set $G_T$
10:    Update policy $\pi_T$ with GCRL
11: **end for**

---

1: **PHASE 2: Learner** ($\pi_L(.|g)$) **training with Teacher's demonstrations**
2: Initialize $\pi_L(.|g)$, goal set $G_L$ to {first goal} and pragmatic Boolean variable
3: **for** $epoch = 1, 2, \ldots$ **do**
4:    Teacher randomly samples goal $g_d$ from goal set $G_L$ and gives demo $d$ by running $\pi_T(.|g_d)$
5:    Learner infers goal $\widehat{g_d}$ of $d$ with BGI
6:    **if** $\widehat{g_d} = g_d$ (feedback provided by the teacher) **then**
7:       Add $d$ to learner's replay buffer
8:    **end if**
9:    Run policy $\pi_L(.|\widehat{g_d})$, obtain trajectory $t = ((s_1, a_1, r_1), .., (s_n, a_n, r_n))$ and achieved goal $g_a$
10:    **if** pragmatic Boolean $= 1$ and $\widehat{g_d} = g_a$ **then**
11:       Infer own goal $\widehat{g_o}$ with BGI on $t$: if $\widehat{g_o} = g_a$, add pragmatic reward to $t$
12:    **end if**
13:    Add $t$ to learner's replay buffer and if $g_a$ is new, add $g_a$ to goal set $G_L$
14:    Update policy $\pi_L$ with GCRL
15: **end for**

---

**Phase 1: Teacher pre-training.** The teacher is pre-trained before providing demonstrations to the learner. The **naive teacher**'s policy is trained to master all goals: it maintains a goal set $G_T$ to which it adds newly encountered goals, it samples goals from $G_T$ and pursues them to collect trajectories added to a replay buffer. Finally, it applies GCRL for policy updates.

The **pedagogical teacher** is also trained to master all goals, but additionally trains itself to predict its own goals using BGI. When training, each time it successfully reaches a goal, it takes its own trajectory and infers the pursued goal using BGI, effectively asking itself: "would a learner be able to predict the goal from this demonstration?". If it correctly infers the goal from its own trajectory, it rewards itself to reinforce its probability to select this trajectory. This is done with GCRL by adding a "pedagogical reward" to the trajectory for which the teacher correctly infers its own goal. This biases policy learning towards finding demonstrations which avoid ambiguity in goal inference, while guaranteeing high performance on the actual task.

**Phase 2: Training the Literal/Pragmatic Learner with Teacher's demonstrations.** The **literal learner**'s goal-conditioned policy $\pi_L(.|g)$ is trained to master all goals in the environment using teacher's demonstrations and GCRL.

At each iteration, the teacher samples a desired goal $g_d$ it wants the learner to achieve. It then presents a demonstration $d$ for the chosen goal to the learner. Using BGI with its own policy, the learner infers the goal $\widehat{g_d}$ from the teacher's demonstration. The teacher provides feedback, telling the learner if $g_d = \widehat{g}$. If the learner correctly inferred the goal of the teacher's demonstration ($\widehat{g_d} = g_d$), the demonstration is added to the learner's replay buffer and used for further training. Then the learner plays its policy conditioned on the predicted goal $\pi_L(.|\widehat{g_d})$ and obtains a trajectory $t$ resulting in an achieved goal $g_a$, which is added to the replay buffer. If $g_a$ is achieved for the first time, then the teacher updates the set of goals from which it can sample. Finally, GCRL improves the policy from its own trajectories and demonstrations drawn from the replay buffer. Note that we assume that the learner does not have access to the goal of the demonstration and thus must infer it, mimicking real-life situations where humans regularly have to infer other people's goals [22].

A **pragmatic learner** improves over a literal learner by training its own policy to better infer the right goals from the teacher's demonstrations. If the learner is able to infer its own goals using BGI, then it will better infer the teacher's goals. The pragmatic learner thus adds a "pragmatic reward" to its trajectories for which it can correctly infer its own goals.

## 4    Experimental Setup

We use two environments as test-beds for our experiments: a simple one inspired from research in Developmental Psychology [22] called "Draw two balls" (DTB) that we initially used for illustrating our approach, and a more complex one called "Fetch block-stacking" (FBS) to test our approach on a more challenging domain [36]. Results on DTB are presented in Appendix A, and the main paper focuses on FBS. Both DTB and FBS are multi-goal environments and present goal ambiguity.

### 4.1    Environment: Fetch Block Stacking

FBS is a block-stacking environment with two Fetch robots (teacher and learner) equipped with robotic arms, see Fig. 2. It is based on MuJoCo [41] and derived from the Fetch tasks [36].

The teacher and learner share goal/state/action spaces. Actions are 4-dimensional: 3D gripper velocities and grasping velocity. Observations are the Cartesian and angular positions and velocities of the gripper and the three blocks. Following [2], we adopt a semantic predicates representation where we add to the observation a binary vector telling whether two blocks are close, and whether any block is on top of any other. The agent uses these binary vectors as goals (with 35 possible goals including stacks and pyramids). Given a particular goal vector configuration, the agent gets a $+1$ reward for each pair of blocks (3 in total) if true predicates about the pair of blocks

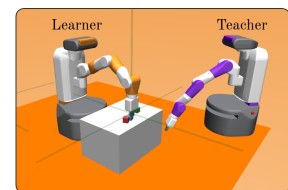

Fig. 2: Fetch Block Stacking.

are the same in the current vector and the goal vector. There are several possibilities to match the true predicates in the goal vector, which induces goal ambiguity. Details about the environment implementation are provided in Appendix B.1.

**Naive/Pedagogical teachers implementation.** The training procedure and policy architecture of the naive teacher are taken from GANGSTR [3] which already implements Fetch Block Stacking. The GANGSTR agent perceives the low-level geometric states and the high-level semantic configurations. Following [3], we encode both object-centered and relational inductive biases in our architecture. We model both the agents policies and critics as Message Passing Graph Neural Networks [19]. We consider a graph of 3 nodes, each representing a single object. All the nodes are interconnected. We consider the agent's body attributes as global features of the policy networks and both the agent's body attributes and the actions as global features of the critic networks.

The agent's goal-conditioned policy is trained using Soft Actor Critic (SAC) [24], a state-of-the-art RL algorithm, combined with Hindsight Experience Replay (HER) [4]. Additionally, the pedagogical teacher rewards itself with the "pedagogical reward" (1 here) when it correctly infers the goal of its successful trajectories. It does so for each of its collected trajectories during training. All architecture, training and hyperparameters details are provided in Appendix B.2.

**Literal/Pragmatic learners implementation.** To implement literal and pragmatic learner training, we use a combination of GANGSTR combined with a slightly modified version of the Soft Q-Imitation Learning (SQIL) algorithm [37]. Originally, SQIL rewards demonstrations to 1 and experience to 0. By contrast, we set the demonstration reward to 1 + the maximum reward obtainable in the environment (3 here), while the reward of trajectories performed in the environment is unchanged. This allows the agent to learn from demonstrations and collected trajectories at the same time using SAC as the GCRL algorithm. As in the original paper, the percentage of demonstrations versus collected trajectories is set to $0.5$. On top of that, the pragmatic learner adds the "pragmatic reward" (1 here) to its trajectories from which it can infer the goal.

### 4.2 Metrics

We use four metrics for all of our experiments: 1) Goal Inference Accuracy (GIA): "is the learner able to correctly infer goals given demonstrations from a teacher?", 2) Own Goal Inference Accuracy (OGIA): "is the agent (teacher or learner) able to correctly infer goals from its own trajectories?", 3) Goal Reaching Accuracy (GRA): "is the learner able to reach all goals in the environment?", and 4) the product of GIA and GRA (GIAxGRA), interpreted as the ability of the learner to predict its goal given a demonstration from the teacher and then reach it. GIA and OGIA help us understand if the pedagogy and pragmatism mechanism actually work, and are computed using a test set of 50 demonstrations per goal generated by the teacher for GIA and by the same agent being tested for OGIA ($50 * 35 = 1750$ demonstrations in total). GRA allows us to evaluate the performance of agents and is computed by testing the agent on each goal 50 times. All demonstrations are long enough to reach the goal (100 timesteps in our case). We provide means $\mu$ and standard deviations over 5 seeds and report statistical significance using a two-tail Welch's t-test with null hypothesis $\mu_1 = \mu_2$, at level $\alpha = 0.05$ (noted by star markers in figures).

Finally, to compare pedagogical and naive demonstrations, we derive an Ambiguity Score: given two ambiguous goals that can be achieved simultaneously and a starting state, the ambiguous score is 1 if their associated demonstrations achieve the same goals (the demonstrations are thus ambiguous), and 0 otherwise. This score assumes that the demonstrator already knows how to achieve the goals. When it is computed from several situations, it evaluates the degree of ambiguity in the demonstrations. For more details on this score please refer to Appendix B.2.6.

## 5 Experiments and Results

We first study learning results from Phase 1 (naive/pedagogical teachers training) and Phase 2 (literal/pragmatic learners trained with teacher's demonstrations). We analyze the impact of demonstrations to accelerate learning, and then scrutinize two algorithmic choices in our approach: goal inference with BGI and learning from demonstrations with SQIL.

### 5.1 Experiments on Phase 1: Naive and Pedagogical Teachers

We verify that the pedagogical teacher can indeed better predict goals from its demonstrations compared to a naive teacher in the FBS environment. Quantitatively, the pedagogical teacher achieves an OGIA of $95.6\% \pm 0.1$ while the naive teacher achieves $83.4\% \pm 0.2$. By training to infer its own goals from its demonstrations, it produces less ambiguous demonstrations. For a qualitative evaluation of this phenomenon, we implement the example in the introduction. We compare demonstrations from a common starting situation (red block close to green block, blue block away) and for two different goals: "Put blue block close to red" and "Put all three blocks close together". As a result, the naive teacher provides the same demonstrations (putting the blue block close to red), while the pedagogical teacher provides two different demonstrations to better disambiguate between the two goals (putting the red block close to blue and putting the blue block close to red and green).

We compute the Ambiguity Score from 500 ambiguous situations sampled from a list of ambiguous situations detailed in Appendix B.2.6 (two ambiguous goals and a starting state) and obtain an unequivocal result: $9\% \pm 1$ for the pedagogical teacher vs $64\% \pm 2$ for the naive teacher. Pedagogical demonstrations are thus 7 times less ambiguous than naive ones.

### 5.2 Experiments on Phase 2: Literal and Pragmatic Learners

We train pragmatic/literal learners with naive/pedagogical teachers and present our results in Fig. 3 and Table 1. In order to evaluate how the mechanisms would respond to training regimes with few demonstrations, we performed the experiments in FBS with 1000, 500 and 100 demonstrations per goal, starting from random states.

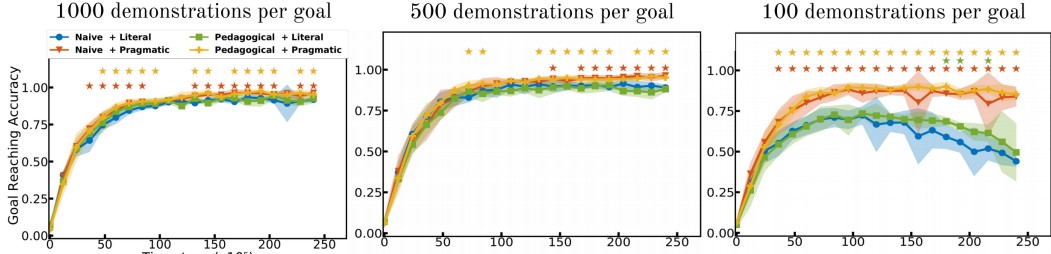

Fig. 3: Global learner performance in the FBS environment (Goal Reaching Accuracy, GRA) with different numbers of demonstrations per goal. Table 1 shows that the drop in GRA under the few demonstrations regime (right) is mainly due to incorrect goal inference in the literal learner or with the naive teacher. Stars indicate significance (tested against naive+literal).

As presented in Fig. 3 and Table 1, it is highly preferable to opt for BGI-agents (a pedagogical teacher with a pragmatic learner). The benefits of adopting pedagogy and pragmatism are particularly prominent in the few demonstrations regime (100 per goal) where pedagogical+pragmatic combination performs 4 times better (going from 0.15 to 0.62) than the naive+literal one in terms of GIAxGRA. Note that with less than 100 demonstrations per goal, none of the approaches are able to master all goals, even if the hierarchy of methods remains unchanged (pedagogy and pragmatism being better than naive and literal, see Fig. 8 in Appendix B.2.7). This is likely due to SQIL forcing a 50/50 split of demonstrations and collected trajectories in the buffer.

An analysis of the pragmatic learner policies reveals that they are better at predicting their own goals, as intended, with an average of 12.4% relative difference in OGIA between a literal and a pragmatic learner. This, coupled with the effectiveness of pedagogical demonstrations shown in the previous section, helps the teacher and learner efficiently communicate and improves learning speed. The benefit of using pragmatism is prominent in the few demonstrations regime: the addition of pragmatism alone (red curve in Fig. 3) helps mastering all goals while pedagogy on its own (green curve in Fig. 3) is not enough to do so. In additional experiments not reported here, we verified that the value of the pedagogical reward and pragmatic reward does not affect performance.

Table 1: Goal Reaching Accuracy (GRA) and Goal Inference Accuracy (GIA) for the FBS environment.

| Teacher + Learner | GIA | GRA | GIAxGRA |
|---|---|---|---|
| With 100 demonstrations per goal | | | |
| Naive + Literal | $30.9 \pm 8.3\%$ | $50.2 \pm 2.7\%$ | 0.15 |
| Naive + Pragmatic | $61.3 \pm 2.9\%$ | $84.8 \pm 4.1\%$ | 0.52 |
| Pedagogical + Literal | $49.6 \pm 6.6\%$ | $62.7 \pm 5.4\%$ | 0.31 |
| Pedagogical + Pragmatic | $\mathbf{69.2 \pm 4.2\%}$ | $\mathbf{89.4 \pm 2.0\%}$ | **0.62** |
| With 500 demonstrations per goal | | | |
| Naive + Literal | $68.9 \pm 1.1\%$ | $90.6 \pm 1.4\%$ | 0.62 |
| Naive + Pragmatic | $76.3 \pm 0.4\%$ | $95.5 \pm 0.6\%$ | 0.73 |
| Pedagogical + Literal | $78.8 \pm 0.7\%$ | $91.5 \pm 1.1\%$ | 0.72 |
| Pedagogical + Pragmatic | $\mathbf{82.2 \pm 1.2\%}$ | $\mathbf{96.8 \pm 0.6\%}$ | **0.80** |
| With 1000 demonstrations per goal | | | |
| Naive + Literal | $75.3 \pm 0.9\%$ | $93.3 \pm 1.2\%$ | 0.70 |
| Naive + Pragmatic | $76.0 \pm 0.9\%$ | $96.6 \pm 0.4\%$ | 0.73 |
| Pedagogical + Literal | $80.0 \pm 1.6\%$ | $94.5 \pm 1.2\%$ | 0.76 |
| Pedagogical + Pragmatic | $\mathbf{84.7 \pm 1.8\%}$ | $\mathbf{97.1 \pm 1.2\%}$ | **0.82** |

### 5.3 Does using the teacher accelerate training for the learner?

In Phase 2 of our setup, the learner is trained using teacher's demonstrations and RL on its own experience. But is the teacher useful? The answer lies in the training speed. We thus compare the training speed of pragmatic learners trained with 100, 500 and 1000 pedagogical demonstrations and their own collected experience (using the same training procedure as before, with SQIL and GANGSTR), and a learner trained with the same architecture and training process, but no demonstration. We end up with a common result in the Learning from Demonstrations literature: the learner trains much faster (here roughly twice faster to obtain a GRA of 90%) when it has access to demonstrations, as illustrated in Fig. 4. This sanity check justifies training learners from demonstrations.

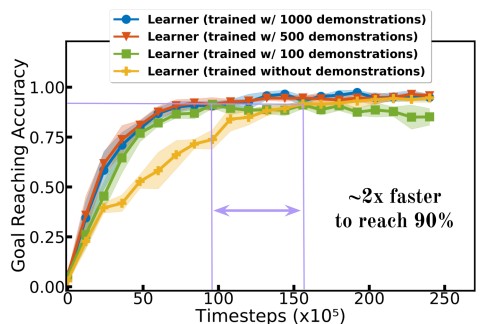

Fig. 4: Demonstrations double the learner's training speed.

### 5.4 Could we use another goal inference method instead of Bayesian Goal Inference?

Table 2: Goal Inference Accuracy (GIA) of BGI and a GPNN on FBS.

| GIA for Pedagogical demonstrations | | | |
|---|---|---|---|
| Nb of demos per goal | 100 demos | 500 demos | 1000 demos |
| GPNN | 46.2% | 70.0% | 82.0% |
| BGI | **69.2**% | **82.2**% | **84.7**% |
| GIA for Naive demonstrations | | | |
| Nb of demos per goal | 100 demos | 500 demos | 1000 demos |
| GPNN | 38.4% | 67.1% | 68.2% |
| BGI | **61.3**% | **76.3**% | **76.0**% |

Our learner relies on BGI using its own policy to infer the goals associated with the demonstrations of the teacher. Could we instead train a separate goal prediction module to learn how to achieve goal inference? We tested this by creating a goal prediction neural network (GPNN) based on a LSTM architecture [30] that inputs a demonstration and outputs a goal, and we trained it using the same demonstrations and goals as for the BGI case. In order to compare our BGI approach to this alternative, we trained GPNN on a different number of demonstrations per goal and compared the results on a separate test set. We performed this experiment on FBS and provide details in Appendix B.2.4. Results from Table 2 show that BGI achieves higher GIA in all cases (naive or pedagogical demonstrations), and again performs especially well in the few demonstrations regime where the GPNN approach struggles because of too few training samples. However, note that in the case of an infinite goal space, a GPNN approach would be viable, while an approach based on BGI would not be straightforward to apply.

### 5.5 What if we used another method to learn from demonstrations?

Our modified version of SQIL helps the learner leverage both demonstrations and the experience it collects. We justify the use of this method by comparing it to baseline approaches: learning only from demonstrations (B1), adding behavioural cloning on the demonstrations (B2) and the original version of SQIL (B3). All implementation

Table 3: Learning from demonstrations baselines.

| Method | GIA (%) | GRA (%) | GIAXGRA |
|---|---|---|---|
| B1 | $5.3 \pm 1.5$ | $6.2 \pm 0.7$ | 0.00 |
| B2 | $2.3 \pm 1.8$ | $52.5 \pm 2.1$ | 0.00 |
| B3 | $59.1 \pm 2.4$ | $73.8 \pm 1.7$ | 0.43 |
| Ours | $\mathbf{69.2 \pm 4.2}$ | $\mathbf{89.4 \pm 2.0}$ | **0.62** |

details are provided in Appendix B.2.5. We perform the comparison using teacher's pedagogical demonstrations and pragmatism in the learner for all methods. B1 is not able to learn, B2 is able to achieve a reasonable GRA score but struggles to predict goals from demonstrations because the behavioural cloning updates interfere with the BGI mechanism by changing the action probabilities. B3 and our modified SQIL approach perform well, with a significant advantage for our approach. The original SQIL was designed for single goal environments and does not provide enough goal-reward associations, while our modification extends the approach to multi-goal environments by providing a reward signal for the collected experience.

## 6 Conclusion

In this paper, building on the pedagogy and pragmatism concepts from Developmental Psychology, we have shown how learning from demonstration can benefit from a Bayesian goal inference mechanism. Using BGI-agents over regular agents improves learning speed, resolves ambiguity in communication and globally improves performance, especially in the few demonstrations regime. In our work, the teacher's demonstration was insensitive to the learner's interpretation, as the teacher did not use a model of the learner's policy to provide a demonstration tailored to that specific learner. Conversely, the learner did not have a model of the teacher's policy which may help it interpret the goal the teacher wanted to convey. In the future, adding in each agent a model of the other agent's policy would result in closing the interaction loop and favoring the emergence of richer interaction mechanisms, such as both partners using signals to help the other update their model of each other during training. Besides, our approach was focused on goal disambiguation for pedagogy and improving goal inference for pragmatism. Pedagogy could be further improved with a curriculum of goals that would improve exploration [3]. Pragmatism could be improved with pedagogy detection [22] and inductive generalization [23]. In the case of multiple teachers, the learner could also develop a mechanism to evaluate teachers and select the most suited one [35]. We leave all this for future work.

Finally, a special trait of our learner is that it learns both from its own signals and from interaction with its teacher, in accordance with the guidelines of [39]. While pedagogy and pragmatism are mechanisms that improve how agents can be taught, they suggest a number of other mechanisms to do so, which in combination to our approach could be beneficial such as language guided learning, internalization, motivation regulation and observational learning.

## Acknowledgements

This work has received funding from European Union's Horizon 2020 ICT-48 research and innovation actions under grant agreements No 952026 (HumanE-AI-Net) and No 765955 (ANIMATAS)

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
