# OpenReview forum: "Pragmatically Learning from Pedagogical Demonstrations in Multi-Goal Environments"
_NeurIPS.cc/2022/Conference — NeurIPS 2022 Accept_

### Official Review · Reviewer_3RPR · 2022-07-06

**Rating:** 6
**Confidence:** 4
**Soundness:** 3 good
**Presentation:** 3 good
**Contribution:** 2 fair

**Summary:**

This work proposes utilizing teacher-learner settings in Goal-conditioned Reinforcement Learning with concepts of pedagogy and pragmatics from other disciplines. In particular, the paper introduces the process of Bayesian Goal Inference (BGI), which is a reward-shaping method that encourages policies to learn behaviors that allow an observer to infer the goal given the trajectory. The paper uses BGI into two settings: to augment a teacher policy’s behaviors to have more disambiguation (“pedagogical teacher”), and to augment a learner to filter our teacher demonstrations and to predict it’s own goals (“pragmatic learner”). The paper applies this method to a toy setting and then a simulated robotic block-pushing setting, and show that pedagogy and pragmatics improve goal reaching and goal inference for both the teacher and the learner.

**Questions:**

- A lot of benefit results from going from literal learner => pragmatic learner, ie. the BGI pedagogical loss for reward shaping. Perhaps the fairest comparison for experiment 4) would be to include cumulative environment interactions (including how many the teacher used). Thus, it can be possible to tell whether the driving benefit results from on-policy updates to encourage the experience-collecting policy (whether it is teacher or learner) to have more disambiguated trajectories. Can you try an experiment where you give the learner (no demonstrations) as many on-policy updates w/ BGI as the cumulative teacher+learner on-policy update budget?
- I could not find information on how much ambiguity results from the environments, as measured by the Ambiguity Score. How percentage of trajectories in the Naive/Literal methods (without BGI) result in ambiguous policies? How does this scale to more complex tasks in higher-dimensional spaces such as learning from vision, where the BGI measure might be much noisier?
- How expensive is the BGI prediction, and how does it scale to continuous action spaces or continuous goal spaces? Given that the method relies on the learner BGI heavily, how does the method fare when goal inference becomes more challenging or a worse method is used? Some possible experiments might be to use GPNN instead of BGI in the main experiments, or an ablation increasing the number of goals.


**Limitations:**

It would be interesting to see how well the method scales to settings where ambiguity is much harder to quantify. In the paper, the authors illustrate the challenges of ambiguity with a three-block toy problem, but in the real world goal inference is often challenging even for experts. In such settings, goal inference auxiliary tasks (like BGI) may not help performance, and may even hurt occasionally.

**Strengths And Weaknesses:**

Strengths:
- The method is quite simple and easily usable with any GCRL + LfD method.
- Since the method implementation in this paper did depend on specific underlying decisions (ie. teacher-learner setup, BGI, SQIL), the ablations were useful to showing the method still works in other settings
- The method increases goal-reaching accuracy in addition to goal-inference accuracy

Weaknesses:
- The method is only tested on a fairly simple block-pushing environment. In more challenging environments (high-dimensional action and observation spaces, more ambiguity in the goal space), it is unclear how noisier BGI predictions will affect the method as it is used for reward-shaping
- Much of the benefit of the method arises from the learner’s utilization of BGI. While 5.4 studies a teacher-less setting, it’s unclear why it’s much more important to have a pragmatic learner than a pedagogical teacher. See questions below.

---

> ### Author Response · Authors · 2022-08-01
> **Response to Reviewer 3RPR: part 1/2**
>
> We thank Reviewer 3RPR for their helpful review and positive assessment of our work. We first address each question, and then go over other weaknesses that the reviewer pointed out.
>
> **Question 1**:  The main point of Sec. 5.3 is to show that sample efficiency is increased when learning with demonstrations and Reinforcement Learning (with SQIL) compared to Reinforcement Learning only – that is, without demonstrations. This ablation is meant to verify that using a teacher is useful. We want to know: if the learner happens to have access to a teacher, is it beneficial for the learner to use this teacher or is it more efficient for the learner to learn on its own? The experiment shows that learning from demonstration helps. This is a common result in the Learning from Demonstrations literature: demonstrations improve sample efficiency. To answer the reviewer’s question regarding the proposed experiment: the reviewer suggested considering the cumulative teacher+learner budget (Budget 1) versus the budget of the learner with no demonstrations (Budget 2). In our case, Budget 1 is twice Budget 2, which would obviously make the learner trained without demonstrations more sample efficient compared to the teacher+learner combination. As far as we know, in the learning from demonstration literature [1], the computational cost of the teacher is not taken into account, which is why we did not count it. Most of the time, the teacher’s policy is even hardcoded or derived from human data. Also, a teacher may train several learners, in which case one should not count the overhead cost for each learner.
>
> **Question 2**: In our work, we do not study whether the policies obtained by the learners are ambiguous or not. Only the policies of the teachers matter: they are the ones that are used to provide demonstrations, which will be naive if there are ambiguities or pedagogical if there are no ambiguities. The story would be different if we were “closing the loop”, that is having the teacher interpreting whether the learner addresses the intended goal from its trajectories, but this richer class of problems is left for future work.
> In a higher dimensional space, e.g. with visual input, other types of ambiguities can emerge. For instance, a “6” seen upside down is a “9” [2], which can create another type of ambiguities (here related to perspective) that have been studied in Human-Robot interaction. Another notorious source of ambiguities is language, which we discuss in our related work section. More generally, the source of possible ambiguities is large and diverse, and we do not claim that our particular implementation of BGI would directly scale to such complex problems. However, the concepts proposed in the paper (making a teacher become pedagogical via own goal inference) can be applied to such problems.
>
> **Question 3**: We thank the reviewer for noticing a limitation of our work we did not consider. The BGI computation comes with a computational cost of performing policy inference, conditioned with all possible goals (here 35). Thus, performing Bayesian Goal Inference over a possibly infinite set of goals doubtlessly raises issues. Our method is not directly applicable to such settings, and modifications are required to cover this case. There has been lots of work on Bayesian Inference in infinite dimensions (see e.g. [5,6]). Another option would be to switch from Bayesian Goal Inference to the baseline called GPNN in the paper (Section 5.4). We thus updated the paper to put forward a relative merit to this alternative in the case of infinite goal spaces. Regarding the possible experiments about adding more goals or experimenting with GPNN instead of BGI: while this is possible in principle, in practice we could not perform those experiments given the 7 days of the rebuttal. However, we agree with the reviewer that these experiments would increase the quality of the paper and we will perform them between the end of the rebuttal and the final notification. If the paper is accepted, we will add the results to the final version, provided that we get a clear message out of them.

---

> > ### Author Response · Authors · 2022-08-01
> > **Response to Reviewer 3RPR: part 2/2**
> >
> > Other weakness/points discussed:
> > - The environment seems relatively simplistic but the agents need to learn to 1) manipulate blocks precisely (a control task), 2) put the blocks in the correct configurations (a goal-oriented task). For a Reinforcement Learning benchmark, this seems sufficiently complex to make our points about pedagogy and pragmatism. Indeed, this type of benchmark has been used extensively in the past in important papers [3,4].
> > - The relative importance of having a pragmatic learner over a pedagogical teacher is only noticed in the few-demonstrations regime in Fig.3, when there are only 100 demonstrations per goal. In this specific regime, the relative importance of pragmatism over pedagogy could be due to the reward shaping scheme that allows the learner to train itself to predict the goals it reaches, and thus less focus on using demonstrations to learn the task.
> >
> >
> > [1]: Argall, B. D., Chernova, S., Veloso, M., & Browning, B. (2009). A survey of robot learning from demonstration. Robotics and autonomous systems, 57(5), 469-483.
> >
> > [2]: Zhao, Xuan, and Bertram F. Malle. "Spontaneous perspective taking toward robots: The unique impact of human-like appearance." Cognition 224 (2022): 105076.
> >
> > [3]: Andrychowicz, Marcin, et al. "Hindsight experience replay." Advances in neural information processing systems 30 (2017).
> >
> > [4]: Nair, Ashvin, et al. "Overcoming exploration in reinforcement learning with demonstrations." 2018 IEEE international conference on robotics and automation (ICRA). IEEE, 2018.
> >
> > [5]: Belitser, Eduard, and Subhashis Ghosal. "Adaptive Bayesian inference on the mean of an infinite-dimensional normal distribution." The Annals of Statistics 31.2 (2003): 536-559.
> >
> > [6]: Hu, Zixi, Zhewei Yao, and Jinglai Li. "On an adaptive preconditioned Crank–Nicolson MCMC algorithm for infinite dimensional Bayesian inference." Journal of Computational Physics 332 (2017): 492-503.

---

### Official Review · Reviewer_4LZc · 2022-07-10

**Rating:** 5
**Confidence:** 3
**Soundness:** 2 fair
**Presentation:** 3 good
**Contribution:** 3 good

**Summary:**

This submission proposes a novel method to learn from demonstration. Inspired by human learning, the authors propose a Bayesian Goal Inference framework to avoid ambiguity in multi-goal learning. And use such a framework to benefit the learner's learning under the demonstrator's help.

**Questions:**

Major concerns:

1. How is the research motivated, why does the pedagogical demonstration and inference of goals important? In GCRL, demonstrations should contain the true goal state, then what is the purpose of inferring multiple potential goals? Real-world applications can be helpful in demonstrating the importance of the studied problem.

2. with those demonstration data, Hindsight Experience Replay methods can be applied to generate more successful goal-reaching experiences. I believe a comparison to HER can be helpful in evaluating the proposed method's performance. (e.g., in Table 3)

3. As for the precise GCRL algorithm the authors used, it was mentioned in line 104 by referring to [11], however, this [11] is a survey paper. More description of the learning algorithm itself should be at least referred to in this section.

4. during the learning of GCRL, the goal is known, then what is the purpose of inferring the goal?

5. the proposed method does not have a theoretical guarantee but is mostly of a shape as an empirical study.

minor:

1. in GCRL (practically, the robotics control tasks, e.g., [36]), the state space and goal space (with a mapping between state and goal space always assumed to be known [36]) are continuous, how will the proposed method tackle such situations where the goal set has an infinite cardinality?

2. Section 4.1 introduces several reward-shaping designs, is the proposed method sensitive to those choices? detailed ablation studies can be helpful to verify the proposed method.

3. 5 seeds are not enough for evaluating RL algorithms.

4. some presentations are ambiguous and need proofreading. e.g., line 100 'where learners are both...', I reckon there is only one learner?

I may misunderstand some parts of the work, and I'm happy to raise my score if the above concerns can be appropriately addressed.

**Limitations:**

1. clear motivation of the research
2. solid guarantee of the proposed method
3. more empirical studies (#seeds and environments)

**Strengths And Weaknesses:**

I appreciate the novelty, and great effort put into the paper. Especially the illustration video in the supplementary material--- which helped a lot in understanding the work.

However, there are several concerns I hope can be addressed according to the current stage of the presentation.
1. motivation/ problem formalism: the authors failed to present the task in a clear enough way: e.g., why does the learner need to infer the goal when the goal is known?
2. missing baselines/ lack of empirical results.
3. lack of theoretical guarantee

---

> ### Author Response · Authors · 2022-08-01
> **Response to Reviewer 4LZc: part 1/2**
>
> We thank Reviewer 4LZc for their insightful review and comments. Below we first provide answers to major points, then to minor points.
>
> Major points:
>
> 1. We thank the reviewer for pointing that the motivation of our work was not clear enough. The motivation is as follows: In real life, when a teacher shows how to do something with a demonstration, the agent receiving the demonstration does not have access to the intended goal of the demonstration: it must infer this goal. In many situations, a single demonstration may be correctly interpreted as demonstration of a variety of goals – we call this goal ambiguity. In that case, the teacher may use pedagogy to help the learner infer the intended goal, and the learner may use pragmatism in order to increase its chance to infer the right goal. This is the situation we address in this paper, with an example given in the first line of the introduction. We have now added this more explicit explanation to the introduction of the revised version of the manuscript.
> 2. All of our agents do use Hindsight Experience Replay in their Reinforcement Learning process. This is specified in Section 4.1. This is also the case for all of our baselines in Table 3.
> 3. We agree that the citation is not correct, we modified it to point to the precise GCRL algorithm we used in our paper. We also added a mention to the name of the algorithm and where to find technical details about it in the updated version of the paper we uploaded. Moreover, we updated Section 4.1 by adding a complete description of how the GCRL algorithm works.
> 4. By inferring the goal of the demonstration provided by the teacher, the learner can match the demonstration with the goal of the demonstration, such that the learner can actually learn from this demonstration. Otherwise, the learner does not know the goal of the demonstration, with the hypothesis we discussed in point 1 (the agent does not have access to the goal of the demonstration, and thus must infer it).
> 5. Our experiments are indeed empirical. Given that we use neural networks to learn to master the goals, we cannot prove for instance that the pedagogical teacher necessarily reduces ambiguities due to mathematical unsolved problems on neural network convergence. However, our empirical evaluation of this phenomenon leaves no doubt about this, see Section 5.1 (Ambiguity Score results). Unfortunately, this shortcoming of not being able to provide theoretical guarantees also applies to most Deep Reinforcement Learning algorithms.

---

> > ### Author Response · Authors · 2022-08-01
> > **Response to Reviewer 4LZc: part 2/2**
> >
> > Minor points:
> > 1. We thank the reviewer for noticing a limitation of our work we did not consider. Indeed, performing Bayesian Goal Inference over an infinite set of goals doubtlessly raises issues. Thus, our method is not directly applicable to such settings, and modifications are required to cover this case. There has been lots of work on Bayesian Inference in infinite dimensions [4,5]. Another option would be to switch from Bayesian Goal Inference to the baseline called GPNN in the paper (Section 5.4). We thus updated the paper to put forward a relative merit to this alternative in the case of infinite goal spaces.
> > 2. The additional rewards provided when activating pedagogy and pragmatism are one of the core elements of our approach. Thus, the ablation study of using the additional pedagogical reward for the teacher is the naive teacher, and the ablation study of using the additional pragmatic reward for the learner is the literal learner. Our results show that they indeed provide a significant boost of performance. We also experimented with different values for these additional rewards and did not see any major change. To make that clear, we mentioned this in the updated manuscript.
> > 3. As the stars in Fig.3 shows, we tested for the statistical significance of our results using the software from [6], which ensures that enough random seeds were used. Nevertheless, to satisfy the reviewer's request, we ran 5 additional seeds in Fig.3 and updated the paper accordingly. The conclusions remain unchanged.
> > 4. We apologize for any ambiguity in our writing. Here the plural was used as a reference to all experiments that were performed in the paper. We clarified this in the paper by changing the sentence to singular. In terms of proof-reading mistakes, we ran the paper in a grammar and spelling checker software to ensure to reduce such errors as much as we can. We would be grateful to the reviewers for pointing out any remaining mistakes if they find any.
> >
> > [1]: Gweon, Hyowon. "Inferential social learning: Cognitive foundations of human social learning and teaching." Trends in Cognitive Sciences 25.10 (2021): 896-910.
> >
> > [2]: Baker, Chris L., Joshua B. Tenenbaum, and Rebecca R. Saxe. "Goal inference as inverse planning." Proceedings of the Annual Meeting of the Cognitive Science Society. Vol. 29. No. 29. 2007.
> >
> > [3]: Dik, Giel, and Henk Aarts. "Behavioral cues to others’ motivation and goal pursuits: The perception of effort facilitates goal inference and contagion." Journal of Experimental Social Psychology 43.5 (2007): 727-737.
> >
> > [4]: Belitser, Eduard, and Subhashis Ghosal. "Adaptive Bayesian inference on the mean of an infinite-dimensional normal distribution." The Annals of Statistics 31.2 (2003): 536-559.
> >
> > [5]: Hu, Zixi, Zhewei Yao, and Jinglai Li. "On an adaptive preconditioned Crank–Nicolson MCMC algorithm for infinite dimensional Bayesian inference." Journal of Computational Physics 332 (2017): 492-503.
> >
> > [6]: Colas, Cédric, Olivier Sigaud, and Pierre-Yves Oudeyer. "A hitchhiker's guide to statistical comparisons of reinforcement learning algorithms." arXiv preprint arXiv:1904.06979 (2019).

---

> > > ### Comment · Reviewer_4LZc · 2022-08-08
> > > **Comments on Author Response**
> > >
> > > Hi Authors,
> > >
> > > I appreciate many of the changes and efforts you've made in the paper to make it clear. I think the idea is now much clearer to me but I'm still not convinced by the setting where we have demonstrations but do not know the goals.
> > >
> > > For general RL tasks, learning from demonstration without knowing the goals seems reasonable, so the IRL research is developed to infer the reward mechanism (as general 'goals'). However, in goal-conditioned R, it seems unnatural for me that we only have demonstrations but don't know the exact goals. In other words, this paper does not work in normal GCRL settings, right?
> > >
> > > Hence, my core question remains: why is it a must to infer the goals in GCRL? Could the authors please explain more?
> > >
> > > I'll raise my score to an acceptance if my concern can be well-addressed.

---

> > > > ### Author Response · Authors · 2022-08-08
> > > > **Response to core concern**
> > > >
> > > > We thank the reviewer for asking for additional clarifications. As we hope our example and explanation have now made clear, the teacher knows what the goal of the demonstration is, but the learner does not. For instance, does the teacher want to demonstrate how to draw a curtain, or to hide the object that is behind?
> > > >
> > > > The goal being not communicated to the learner, the learner cannot fill the "goal part" of a goal-conditioned policy as it would do when using standard goal-conditioned learning from demonstration, and the learner needs to infer the goal to appropriately fill this goal part.
> > > >
> > > > **So yes, we are using GCRL, but we further assume that the teacher does not communicate the goal when performing the demonstration, which results in a more complicated setting.** We believe this setting to be highly relevant in future scenarios where human users will teach robots from demonstration with robots that cannot interpret language, as in such cases the user will not be able to directly communicate the goal while performing the demonstration.
> > > >
> > > > We hope this further clarification will help the reviewer, otherwise please keep this conversation running as we believe we can reach mutual understanding before the end of the discussion period.

---

> > > > > ### Author Response · Authors · 2022-08-09
> > > > > **Additional response to main concern**
> > > > >
> > > > > As the deadline for the end of the discussion period approaches, we would like to add an argument that may clarify the concern.
> > > > >
> > > > > In real-world problems, inferring goals from demonstrations and more generally actions is a crucial part to exchange information and learn efficiently, as a consensus of work in developmental psychology and machine learning shows [1, 2, 3]. Indeed, most of the time agents do not communicate their goals explicitly when they act [1] and agents have to infer the goals of other agents. Humans, and even infants, particularly excel at inferring goals when observing the behavior of other persons. This allows them to understand their intention, how to achieve goals later on their own, and is generally a powerful cognitive mechanism for their development and learning. In our case, the agent receiving the demonstration does not have access to the goal of the demonstration, in order to mimic what would happen in real-life scenarios.
> > > > >
> > > > > In order to best satisfy the reviewer, we updated the manuscript to better state the assumption that the learner does not have access to the goals of the demonstrations (and thus must infer it). In our description of the teacher/learner setup in the context of GCRL (Phase 2 in Sec.3.2), we now clearly state this assumption:  "Note that we assume that the learner does not have access to the goal of the demonstration and thus must infer it, mimicking real-life situations where humans regularly have to infer other people's goals [1]".
> > > > >
> > > > > We hope that these arguments will help the reviewer understand the importance of goal inference in communication between agents, and thus our motivation to apply to GCRL.
> > > > >
> > > > >
> > > > > [1]: Gweon, Hyowon. "Inferential social learning: Cognitive foundations of human social learning and teaching." Trends in Cognitive Sciences 25.10 (2021): 896-910.
> > > > > [2]: Baker, Chris L., Joshua B. Tenenbaum, and Rebecca R. Saxe. "Goal inference as inverse planning." Proceedings of the Annual Meeting of the Cognitive Science Society. Vol. 29. No. 29. 2007.
> > > > > [3]: Dik, Giel, and Henk Aarts. "Behavioral cues to others’ motivation and goal pursuits: The perception of effort facilitates goal inference and contagion." Journal of Experimental Social Psychology 43.5 (2007): 727-737.

---

> ### Author Response · Authors · 2022-08-08
> **Quick follow up**
>
> Thank you again for your review. We wanted to quickly follow up and see if our response adequately addresses your questions and comments, or if you have additional concerns.

---

### Official Review · Reviewer_yuXv · 2022-07-10

**Rating:** 5
**Confidence:** 4
**Soundness:** 3 good
**Presentation:** 2 fair
**Contribution:** 2 fair

**Summary:**

* This work focuses on the problem of learning from teacher/learner demonstrations and in particular focus on training a teacher/learner setup where goal ambiguity can be resolved to improve the quality of the demonstrations and to thus improve effective goal inference and task performance.  Multi-goal environments are utilized.
* The authors make use of concepts from cognitive science, pedagogy: optimization of teaching concepts; Pragmatism: resolution of ambiguities of intention from the teacher.
* With this in mind the authors aim to tackle the problem of goal ambiguity: given that at least two goals are reachable in a given trajectory can reach at least two goals -> can't reliably predict pursued goal.  The authors aim to improve demonstrations by resolving goal ambiguity and learning good goal inference.
* Teacher/Learner share common goal / state / action spaces and use separate policies.  Communication between the two is facilitated via teacher demos, learner's inferred goal, teacher feedback on inference.

Training proceeds roughly as follows:

1. Teacher is pre-trained in the env to master/solve all goals with Goal Conditioned RL (GCRL)
2. Learner infers goal from teacher's demo using Bayesian goal inference (BGI), teacher provides feedback, learner rewarded for correct predictions

The teacher tries to infer its own goal (with BGI) from trajectory after reaching it.  It adds a "pedagogical" reward to the trajectory if so which aids in retaining high performance on the task.

Results measured over the Draw-Two-Balls (DTB) and Fetch-Block-Stacking (FBS) environments.  Four metrics are collected:

1. Goal inference accuracy (GIA) - learner can infer G from d
2. Own goal inference acc - teacher can infer G from its own demonstrations
3. Goal reaching acc (GRA)  - can learner reach all goals?
4. GIA x GRA - can the learner both infer its goal and reach it
Ambiguity Score

The main claim of the paper is that using a pedagogical and pragmatic teacher/learner setup to improve demonstrations through goal disambiguation will lead to to faster learning with fewer demonstrations.


**Questions:**


* How can we ensure that all goals are mastered/solved/achieved by the teacher over the dataset?  What happens if we are only able to master some fraction?
* In alg. 1 are flags provided for "learner is pragmatic" and "teacher is pedagogical"?
* What do you think causes the accuracy dip for the learner curve with no demonstrations in figure 1?
* What computational overhead does the pre-training occur?  How does this compare to the phase involving learning from the demonstrations?


**Limitations:**

Limitations weren't explicitly addressed in the main paper to my knowledge.

**Strengths And Weaknesses:**

**Strengths**:

* The authors aim to tackle an interesting problem of leveraging teach/learner demonstrations which are important in a variety of domains.  * The approach they take to disambiguate goals from existing demonstrations is very compelling and could lead to learning improvements in a variety of ways (signal boost, sample efficiency, data quality)
* Overall the approach presented in the paper is sound and the paper is clearly written for the most part although there may have been a bit more in the way of architectural detail.
* The authors provide useful context for their work in section 2 describing past relevant work in pedagogical demos, pragmatic inference and demonstrations in goal conditioned tasks.
* The training phases are detailed well with accompanying algorithms.
* The authors show that in the context of naive/simple v. pedagogical/pragmatic teacher/learner setups the pedagogical+pragmatic variant has a clear advantage in the evaluated tasks.


**Weaknesses**:

* In the  methods section some more explanation on how new goals are discovered could be helpful.  I'm a bit foggy on how the goals are actually selected.
* In section 4.1: A bit more explanation around the distinction of literal v pragmatic learners and naive v pedagogic teachers would be helpful.  Some basic details in the main paper would be helpful.  Same goes for the architectural details of the teacher/learner networks.  The training and interaction is explained however there are no clear architecture diagrams and not much explanation in the main paper.
* Results in the main paper are limited to a single task and while they show better sample efficiency in this case it is hard to understand the benefit realized more generally.  Further the 95% threshold for the 2x increase claimed seems somewhat hand picked (e.g. why not 80% or 90%?  These would reduce the sample efficiency significantly).
* From the results it seems clear that Pedagogical/Pragmatic Teacher/Learner improve over Naive/Simple variants however it is less clear how well these improve more generally over other approaches.  Section 5.5 seems to address this over a limited set of baselines including Behavioural Cloning and SQIL but these don't seem to be particularly strong baselines (a couple of them don't even reach a substantial number of goals in the demos).  I believe more rigorous comparisons to baselines would be needed here in order to really justify the claims of this work.
 * In figure 4 there is a notable dip over the learner without demonstrations at roughly 6e5 learner steps.  It would be useful to explain what this might be, especially since it seems to be present for all seeds and only for this condition. It would be helpful to understand these gains in a broader context to understand the full benefit of the work here.  It also may have been useful to try with fewer than 100 demonstrations as performance seems to be mostly unaffected when reducing the number of demos from 1000.
* There are no results for DTS results in the main paper, including these could be useful.

Overall I think that this is a promising are of work but some more detail would help to explain the approach and the experiments/results would benefit from being more comprehensive to satisfy the claims.


[2022-07-27] EDIT:  I've added in the missing bit the authors referenced in their comment (I can't reply Officially yet, so adding here).  Apologies for that and thanks for flagging.

---

> ### Author Response · Authors · 2022-08-01
> **Response to Reviewer yuXv: part 1/2**
>
> We thank Reviewer yuXv for their thorough and helpful review. We first answer questions and then go over all other weaknesses raised in the review.
>
> **Question 1**: The reviewer wonders what happens if we cannot ensure that all goals are mastered/solved/achieved by the teacher over the dataset. In our work, we need to assume 100% mastery of goals from the teacher, and we train it until it reaches a Goal Reaching Accuracy close to 100%. If the teacher is not able to master a goal, then its demonstrations for this goal are not helpful, and since the learner and teacher share the same architecture and learning process for exploration, there is no hope of having the learner master that particular goal if the teacher cannot.
>
> **Question 2**: The reviewer rightfully points to two missing flags in Alg. 1. Indeed, the flags “learner is pragmatic" and "teacher is pedagogical” refer to two booleans that are true if the learner is pragmatic (respectively if the teacher is pedagogical) and false if the learner is literal (respectively if the teacher is naïve). These two boolean numbers correspond to “pragmatic_learner” and “pedagogical_teacher” variables in the codebase provided in the supplementary. We added these variables in the initialisation phase of Phases 1 and 2 in Alg.1 and clarified the sentences “learner is pragmatic” and “teacher is pedagogical” to refer to these variables. We thank the reviewer for having spotted this missing information.
>
> **Question 3**: About the dip for the learner in Fig. 4, we thank the reviewer for catching an error we made in the plot. Indeed, there was a bug in the data used for the plot in Figure 4. Only one seed was used instead to compute the mean for the yellow curve (learner trained without demonstrations), and the error bars were erroneous. We should have seen that each learning curve should be monotonically increasing. We rectified this error in the updated manuscript by plotting the correct data. The curve now makes sense, and does not dip anymore. Regarding the point of this experiment, the conclusions remain the same. The main point of Sec. 5.3 is to show that sample efficiency is increased when learning with demonstrations and Reinforcement Learning (with SQIL) compared to learning without demonstrations and Reinforcement Learning only. This ablation is meant to verify that using a teacher is useful.
>
> **Question 4**: About the computational overhead from the pre-training process, it is the same as the computational cost of training the learner without demonstrations. In a setup with a teacher teaching a learner to master goals using demonstrations, we do not consider the computational cost of training the teacher. As far as we know, in the learning from demonstration literature [1], the computational cost of the teacher is not taken into account, which is why we did not count it. Most of the time, the teacher policy is even hardcoded or derived from human data. In Section 5.3, the goal is to show that learning with demonstrations is superior to learning without demonstrations, which verifies that our learning setup indeed benefits from using demonstrations. The rest of the paper involves showing that adding pedagogy and pragmatism is beneficial too.

---

> > ### Author Response · Authors · 2022-08-01
> > **Response to Reviewer yuXv: part 2/2**
> >
> > Other weakness/points discussed:
> >
> > - New goals are discovered by the learner through exploration while pursuing a goal. Indeed, the Reinforcement Learning algorithm (SAC) leverages exploration to occasionally discover new goals. These new goals can then be demonstrated by the teacher, such that the learner can master them. This is specified in Alg.1, and in Phases 1 and 2 in Section 3.2. Regarding how goals are selected, the teacher randomly selects goals to demonstrate among the goals discovered by the learner. We added the mention of randomness in goal selection in Alg.1 to clarify this point.
> > - We would love to include more details about architectures, training (which are provided in full details in the appendix) in the main paper. Unfortunately we clearly lack space. However, if the paper is accepted, we will gain an additional page, which we will use to add those details by simply moving them from the appendix to the main paper. We already prepared this novel version of the paper, but we were informed by the Program Chairs that we cannot upload a 10-page version of the paper during the rebuttal.
> > - Our results show increased sample efficiency when using pedagogy and pragmatism (resulting from the reduced amount of goal ambiguity in the demonstrations). Sample efficiency is one benefit of improving goal communication between the teacher and the learner, but we infer other benefits that could arise in different setups: if the goal is miscommunicated, there might be catastrophic consequences in real-life scenarios (grabbing a cold plate versus grabbing a very hot plate while cooking for instance). About the choice of 95% threshold, it is indeed arbitrary, we just wanted to illustrate the difference in sample efficiency, see the answer to Question 3.
> > - Regarding insufficient baselines, the baseline considered in Section 5.5 are there to justify the use of our modified version of SQIL instead of another learning from demonstration method. We simply show that even with pedagogy and pragmatism, the three baselines considered (learning from demonstrations only, behavioral cloning and the original version of SQIL) do not reach a sufficient Goal Reaching Accuracy to master all goals in our teacher/learner experimental setup. Thus, it is legitimate to base our results on our modified version of SQIL, at least for this environment (Fetch Block Stacking). If we agree on the fact that pedagogical/pragmatic combination clearly improves upon naive/literal combination, then we agree on the main point of our paper.
> > - Regarding experimenting with less than 100 demonstrations per goal, we agree with the reviewer’s comment, which led us to perform the experiment of Fig.3 with 10 demonstrations per goal. We obtained the following results: the hierarchy of methods is respected (pedagogical + pragmatic > naive + literal), but none of the approaches were able to master all goals. This is certainly due to the SQIL method we use, which forces a 50/50 split of demonstrations versus collected trajectories by the learner in the buffer of the Reinforcement Learning algorithm. Hence, this does not change our main claim around pedagogy and pragmatism being helpful when training with demonstrations. We added the figure in the appendix, and added a comment about those results in the main paper.
> > - Results about DTS can be moved in the main paper, at least shortly, if the paper is accepted given the additional page that we will get. We already prepared this novel version of the paper, but we were informed by the Program Chairs that we cannot upload a 10-page version of the paper during the rebuttal.
> >
> > [1]: Argall, B. D., Chernova, S., Veloso, M., & Browning, B. (2009). A survey of robot learning from demonstration. Robotics and autonomous systems, 57(5), 469-483.

---

> > > ### Comment · Reviewer_yuXv · 2022-08-08
> > > **Response to Authors**
> > >
> > >
> > > Thank you very much for the extra experiments and the added clarity in the paper, it's resolved some of the issues I had from my end and improved the presentation  My biggest concerns are still over the scale of the contribution that follows from the results and whether this is clear to the reader.
> > >
> > > Mainly, what I'd like to see is a clearer demonstration of how the pedagogical+pragmatic teacher/learners can be expected to lead better goal finding agents generally.  The authors comment:
> > >
> > > > Sample efficiency is one benefit of improving goal communication between the teacher and the learner, but we infer other benefits that
> > > > could arise in different setups: ... [example of picking up a hot and cold plate]
> > >
> > > Would it be possible to demonstrate such an example in the data?  In general, the current set of results seems mainly to demonstrate that the pedagogical demonstrations with goal inference seems to improve sample efficiency but not necessarily to improve overall accuracy or only to do so incrementally.  This is an important result but I believe if the strength of the contribution is to stand on that it should be better explained why this is an important result (e.g. due to the cost of obtaining demos in these domains) or even better to show some set of tasks or environments where this approach leads to an overall performance benefit via this type of goal inference.  I think in general the results section could use improvement in this way in order to strengthen the contribution and better contextualize the work.
> > >
> > > In summary, I'm on the fence and I do like this approach but I think the results need to better demonstrate the importance of the claims.

---

> > > > ### Author Response · Authors · 2022-08-09
> > > > **Response to Reviewer yuXv**
> > > >
> > > > We warmly thank the reviewer for constructively asking us to better put forward our most important results so as to make the paper more convincing. First, the reviewer expresses the concern that  the current set of results may only "demonstrate that the pedagogical demonstrations with goal inference improves sample efficiency but not necessarily overall accuracy or only does so incrementally". On that specific point, we agree that only looking at Fig. 3 p. 8, the reader may not be able to sort out whether the gain is in sample efficiency or goal accuracy: on the low data regime (100 demonstrations) we see that the literal learner fails to reach more than 60% of goals, but is it because it wrongly inferred the goals or because its policy is not trained enough? The answer to that is in Table 1, p. 7. By sorting out GIA (Goal Inference Accuracy) and GRA (Goal Reaching Accuracy), we can see that the literal learner mainly fails because it wrongly infers the goal.
> > > >
> > > > To better stress this point, we have edited the caption of Fig.3.
> > > >
> > > > Instead of "Results for FBS environment (Goal Reaching Accuracy (GRA) with different numbers of demonstrations per goal). Stars indicate significance (tested against naive+literal).",
> > > >
> > > > we now have:
> > > >
> > > > "Global learner performance in the FBS environment (Goal Reaching Accuracy, GRA) with different numbers of demonstrations per goal. Table 1 shows that the drop in GRA under the few demonstrations regime (right) is mainly due to incorrect goal inference in the literal learner or with the naive teacher. Stars indicate significance (tested against naive+literal)."
> > > >
> > > > We hope that this change will better highlight the outcome of our empirical study.
> > > >
> > > > Beyond this specific point, the reviewer is advising us to better stress the importance of our results. Actually, without calling upon a more complicated example, with Fig. 3 we already have an example where the literal learner paired to the naive teacher succeeds much less than the pragmatic + pedagogical pair. In the low demonstration regime, imagine that when you teach your robot, it fails 50% of the time versus 10% of the time with the pragmatic + pedagogical pair. That already makes a concrete and significant difference.
> > > >
> > > > Due to page limits (we already had to edit some other text so that our additions fit in), we could not add a whole paragraph to better stress this point, but we slightly edited the conclusion (mentioning the increase in global performance, see p.9) to make the point clearer. If the paper is accepted and benefits for an additional page, we can add the following sentence if the reviewers agree on that:
> > > >
> > > > "In real world applications such as teaching a robot where human demonstrations are costly, the pair of pedagogical+pragmatic inference mechanisms may increase the learner's performance by several tens of percent, which can make a real difference."
> > > >
> > > > We hope we have answered the reviewer's concerns and we are ready to discuss further if anything else needs to be raised.

---

> ### Author Response · Authors · 2022-08-08
> **Quick follow up**
>
> Thank you again for your review. We wanted to quickly follow up and see if our response adequately addresses your questions and comments, or if
> you have additional concerns.

---

### Official Review · Reviewer_ozr3 · 2022-07-15

**Rating:** 8
**Confidence:** 4
**Soundness:** 4 excellent
**Presentation:** 4 excellent
**Contribution:** 4 excellent

**Summary:**

This paper leverages the ideas of pedagogy and pragmatism for training goal-conditioned RL agents in multi-goal environments with demonstrations. Pedagogy refers to the ability of the teacher to provide non-ambiguous demonstrations to the learner, whereas pragmatism refers to the learner’s ability to discern which goal the teacher intended to demonstrate. The algorithm consists of two phases: 1) a goal-conditioned teacher policy is trained using off-policy trajectories. In addition to the reward from the environment, the teacher policy receives a reward for correctly inferring the intended goal of its own trajectory using Bayesian Goal Inference (BGI). 2) The teacher provides demonstrations to a new learner policy, which receives rewards from the environment and additional rewards for correctly inferring the intended goal of the teacher's demonstrations, and the intended goal of its own trajectories. The experimental results show that pedagogical teachers and pragmatic leaners are useful when learning from demonstrations.

**Questions:**

* Is the teacher's ability to master all goals in the environment a requirement for the use of this algorithm? Were any experiments done to see if training both the student and teacher in a cooperative manner had better results (essentially combining phases 1&2 together)?
* How is the first goal chosen in phases 1 and 2? Could strategically choosing the first goal impact the performance of the teacher and student?
* Do the authors think that increasing the number of demonstrations would make GPNN perform better compared with BGI?

**Limitations:**

The limitations were well addressed in Section 6.

**Strengths And Weaknesses:**

### Originality
The related work section is comprehensive, and I appreciate the authors breaking the different areas into different sections. This work seems to be a novel combination of pedagogy, pragmatism, BGI, and goal-conditioned RL in multi-goal environments.

### Quality
The experimental results were presented well and clearly addressed the contributions stated in the introduction. The authors do a great job of including specific information such as architecture, number of random seeds, mean, variance, etc.

The questions that the experiments aim to answer seem to be 1) can the learner infer goals from the teacher, 2) can both the learner/teacher infer goals from their own trajectories, 3) can the learner reach all goals, and 4) can the learner both predict goals and reach them.

$GIA$ addresses 1, $OGIA$ addresses 2, $GRA$ addresses 3, and $GIA \times GRA$ addresses 4. The results for $GIA$, $GRA$, and $GIA \times GRA$ are included in Table 1, but I did not see the results for $OGIA$?

### Clarity & Significance
Overall, the paper is very well written and easy to follow. The authors motivated the work in an intuitive way, and also highlighted potential applications. The results seem to be significant, especially for areas like robotics where acquiring demonstrations can be expensive. Researchers can easily extend this approach to use pre-existing policies for the teachers, then further refine them to be pedagogical.

* In Section 5.1, it is implied that the experiments are referring to Fetch Block Stacking, but I think this should be stated in the first sentence for clarity. E.g., "We verify that the pedagogical teacher can indeed better predict goals from its demonstrations compared to a naive teacher 'in the FBS environment'".
* I don't completely understand  how the Draw Two Balls environment is defined. If there are purple, orange, and pink balls, and two consecutive draws, how are there only three possible outcomes?
* Nits
    * I think the second line of Figure 5 should say "goal 2 (activation of sound 2)"
    * Line 85: remove the comma
    * Line 119: add comma "Finally,"

---

> ### Author Response · Authors · 2022-08-01
> **Response to Reviewer ozr3: part 1/2**
>
> We thank Reviewer ozr3 for their helpful review and for their very positive appreciation of our work. We have treated the minor mistakes pointed out by the reviewer and we answer their main questions below.
>
> **Question 1**: The reviewer wonders whether having the teacher mastering all goals is mandatory in our work. In the scope of the paper, it is: we only consider the case where the teacher masters all goals before trying to teach anything to the learner, which is the classical setup in Learning from Demonstrations [1]. Having the teacher learning to reach some goals simultaneously with teaching would make our work closer to Asymmetric Self Play [2], but this would raise additional issues that we do not consider here.
>
> **Question 2**: The reviewer wants to know more about the goal selection strategy. In this work we consider a random goal selection among discovered goals, as having a curriculum over goals is not our focus. In practice, the first goal is the goal representing all blocks far apart, which any agent can achieve because it is the starting state in the environment. This goal acts as a placeholder for the goal-conditioned policy so that the agent starts from scratch. Then, the agent can actually discover new goals via exploration. If the first goal was chosen strategically (two cubes next to each other for instance), then the teacher would sample this goal more often at the beginning of training, and the learner would eventually get demonstrations for this goal early on during training. This might increase the learner’s performance. Indeed, strategically choosing which goal to follow and demonstrate first is a way to provide a curriculum of goals. Providing a curriculum of goals would impact the learner's performance, but would not impact the ambiguity of demonstrations, and is thus orthogonal to the concepts of pedagogy and pragmatism which are the core interests in our work. To conclude, introducing a curriculum of goals (which might be a careful selection of increasingly harder goals to provide a scaffolding strategy) could be an addition to our work in the future, but is not our focus here. Notably, many works related to agents learning from demonstrations provide goals which are in the “Zone of Proximal Development” (a concept derived from developmental psychology [3]), i.e. neither too easy nor too hard for the agent to learn [4,5].
>
> **Question 3**: The reviewer wants to know whether increasing the number of demonstrations would make GPNN perform better compared with BGI. The question being legitimate, we ran the experiments to provide an answer based on actual data. We trained a GPNN on 5000 demonstrations per goal, for both naive demonstrations and pedagogical demonstrations. We obtained the following results in terms of GIA: 75.6% for naive demonstrations (compared to 76.0% for BGI) and 87.4% for pedagogical demonstrations (compared to 84.7% for BGI). Hence, with 5000 demonstrations, GPNN scores on par with BGI for naive demonstration, whereas it marginally surpasses it for pedagogical demonstrations. The conclusion from this experiment remains untouched, BGI remains the right option when there are few demonstrations. Note that 5000 demonstrations per goal equals 175k total unique demonstrations, which is a lot considering that most Learning from Demonstrations methods try to maximally reduce the number of demonstrations used [6,7].

---

> > ### Author Response · Authors · 2022-08-01
> > **Response to Reviewer ozr3: part 2/2**
> >
> > Other weakness/points:
> >
> > - Results about OGIA: these results are not provided in tables or figures but rather in the text. You can find the results in Section 5.1 for the teacher (95.6% for pedagogical teachers and 83.4% for naïve teachers), and Section 5.2 for learners (average of 12.4% increase in OGIA for pragmatic learners over literal learners).
> >
> > - We added “in the FBS environment” in the beginning of Section 5.1, please refer to the uploaded new version of the paper.
> >
> > - In the DTB environment, there are three outcomes (pairs of balls) that achieve goals, out of the 2^3=8 outcomes. Indeed, the sentence is confusing, so we corrected it in the updated version of the manuscript.
> >
> > [1]: Argall, B. D., Chernova, S., Veloso, M., & Browning, B. (2009). A survey of robot learning from demonstration. Robotics and autonomous systems, 57(5), 469-483.
> >
> > [2]: Sukhbaatar, S., Lin, Z., Kostrikov, I., Synnaeve, G., Szlam, A., & Fergus, R. (2017). Intrinsic motivation and automatic curricula via asymmetric self-play. arXiv preprint arXiv:1703.05407.
> >
> > [3]: Cole, Michael. "The zone of proximal development: where culture and cognition." Culture, communication, and cognition: Vygotskian perspectives 146 (1986).
> >
> > [4]: Akakzia, Ahmed, et al. "Grounding language to autonomously-acquired skills via goal generation." arXiv preprint arXiv:2006.07185 (2020).
> >
> > [5]: Fournier, Pierre, et al. "Clic: Curriculum learning and imitation for object control in non rewarding environments." IEEE Transactions on Cognitive and Developmental Systems 13.2 (2019): 239-248.
> >
> > [6]: Duan, Yan, et al. "One-shot imitation learning." Advances in neural information processing systems 30 (2017).
> >
> > [7]: Kim, Beomjoon, et al. "Learning from limited demonstrations." Advances in Neural Information Processing Systems 26 (2013).

---

> > > ### Comment · Reviewer_ozr3 · 2022-08-03
> > > **Response to Authors**
> > >
> > > Thank you for addressing all of my comments and taking the time to run additional experiments! I think I will keep my score at 8 given the criteria for 9 and 10.
> > > ***
> > > Additional Questions:
> > > * Would it be possible to add OGIA results to Table 2 for each ablation?
> > > *  I'm still a bit confused by the DTB environment example in Figure 5. Goal 1 is (orange, orange) and Goal 2 is (pink, orange). How does (orange, pink) activate both Goals 1 and 2?

---

> > > > ### Author Response · Authors · 2022-08-04
> > > > **Response to additional questions**
> > > >
> > > > Regarding the additional questions:
> > > >
> > > > - If the paper is accepted, we will get an additional page, which we can use to add the OGIA results in Table 2. For now we lack space, and the program chairs explicitly told us that we cannot go over the 9 pages limit during the rebuttal.
> > > >
> > > > - About the DTB environment, here is a clarification: there are three ways to reach goal 1 ((orange, orange), (pink, orange) and (orange, pink)). This goal is characterized by a sound (let's say sound 1). Additionally, there is only one way to reach goal 2 ((orange, pink)) and this goal is characterized by sound 2. This means that doing (orange, pink) allows the agent to reach both goal 1 and 2 (both sound 1 and 2 will be played). This is the source of ambiguity which will make a naive teacher confusing for a learner, and that the pedagogical teacher will avoid (by avoiding the demonstration (orange, pink) when demonstrating goal 1 for instance).
> > > >
> > > > We thank the reviewer again for their helpful review.

---

> > > > > ### Comment · Reviewer_ozr3 · 2022-08-06
> > > > > **Response to Authors (8/6)**
> > > > >
> > > > > Thanks, I understand the goal ambiguity of the DTB environment now!

---

### Author Response · Authors · 2022-08-01
**General response to all reviewers.**

We thank all reviewers for providing a thorough evaluation of our work. Their constructive remarks have helped us improve the quality of our work, as can be seen in the revised version that we just uploaded. Our response to each reviewer below is self-contained, but here we provide a general answer to comments that were found in multiple reviews.

1) We updated the manuscript with several modifications thanks to the reviewer's comment. They all appear in red.

2) Two reviewers raised a point about counting the training of the teacher as an overhead of computation cost regarding the experiments in Section 5.3 (Figure 4). As far as we know, in the learning from demonstration literature [1], the computational cost of the teacher is not taken into account, which is why we did not count it. Most of the time, the teacher policy is even hardcoded or derived from human data. In Section 5.3, the goal is to show that learning with demonstrations is superior to learning without demonstrations, which verifies that our learning setup indeed benefits from using demonstrations.

3) Two reviewers also pointed out that our method might not scale to an experimental setup with a greater number of goals, and possibly an infinite amount of goals. We thank the reviewers for noticing a limitation of our work we did not consider. Indeed, performing Bayesian Goal Inference over an infinite set of goals doubtlessly raises issues. Thus, our method is not directly applicable to such settings, and modifications are required to cover this case. There has been lots of work on Bayesian Inference in infinite dimensions (see e.g. [1,2]). Another option would be to switch from Bayesian Goal Inference to the baseline called GPNN in the paper (Section 5.4). We thus updated the paper to put forward a relative merit to this alternative in the case of infinite goal spaces.

4) We rectified a number of other important points in the paper, which we list here:

    - There was an error on Figure 4 regarding the plotted data, which caused a dip in performance of the learner without demonstrations. The data used for the plot was not correct. We corrected it, and the conclusions of the experiments remain unchanged. We thank Reviewer yuXv for noticing this error.
    - We clarified Alg.1 by providing details about the pedagogical and pragmatic boolean variables used in the algorithm, as well as the random goal selection process among discovered goals. Upon acceptance, the paper can be 1 page longer, so in order to fulfill the reviewers' requests we will update the paper with content that is currently in the appendix: 1) details about architecture and training, 2) a summary of results on the Draw Two Balls environment.
    - There was a concern on the number of seeds used in the main experiments (Section 5.1 and 5.2). As the stars in Fig.3 show, we tested for the statistical significance of our results using the software from [3], which ensures that enough random seeds were used. Nevertheless, to satisfy the reviewer's request, we ran 5 additional seeds in Fig.3 and updated the paper accordingly. The conclusions remain unchanged.
    - There was some concern that the GPNN baseline would significantly outperform our BGI method if we were to use more demonstrations in Section 5.4. We ran additional experiments with 5000 demonstrations per goal and showed that GPNN does not significantly outperform BGI.

We again thank the reviewers for their work. If any reviewer feels that our answers have not correctly addressed their point, we would be delighted to answer a further set of questions. If the reviewers feel that our answers are satisfactory, we would appreciate if they would consider raising their score.

[1]: Belitser, Eduard, and Subhashis Ghosal. "Adaptive Bayesian inference on the mean of an infinite-dimensional normal distribution." The Annals of Statistics 31.2 (2003): 536-559.

[2]: Hu, Zixi, Zhewei Yao, and Jinglai Li. "On an adaptive preconditioned Crank–Nicolson MCMC algorithm for infinite dimensional Bayesian inference." Journal of Computational Physics 332 (2017): 492-503.

[3]: Colas, Cédric, Olivier Sigaud, and Pierre-Yves Oudeyer. "A hitchhiker's guide to statistical comparisons of reinforcement learning algorithms." arXiv preprint arXiv:1904.06979 (2019)

---

### Meta-Review · Area_Chair_2fKw · 2022-08-25

**Recommendation:** Accept
**Confidence:** Certain

**Metareview:**

After a strong rebuttal from the authors and an extensive discussion among the reviewers, I believe the paper's pros outweigh its cons and this paper will be a valuable contribution to NeurIPS. I recommend it for acceptance and encourage the authors to address the reviewers comments for the camera-ready version of the paper, especially regarding the weaknesses of empirical evaluation and differentiation against conventional GCRL.


**Award:**

No

---

### Decision · Program_Chairs · 2022-09-14

Accept